# Cost-sharing and associated factors in the Peruvian private health care system

**Luciana Bellido-Boza**[1,2‡]*, **David Villarreal-Zegarra**[1,3‡], **Max Pariona-Cárdenas**[2], **Roger Carrión**[2], **Paul Valdivia-Miranda**[2], **G. J. Melendez-Torres**[4]

**1** Facultad de Ciencias de la Salud, Universidad Peruana de Ciencias Aplicadas, Lima, Peru,
**2** Superintendencia Nacional de Salud, Lima, Peru, **3** Instituto Peruano de Orientación Psicológica, Lima, Peru, **4** Faculty of Health and Life Sciences, University of Exeter, Exeter, United Kingdom

‡ LBB and DVZ share the first authorship on this work.
* lubellido@gmail.com

**Data Availability Statement:** This research utilized the TEDEF-SUSALUD database, which is freely accessible on the open data platform of SUSALUD. The database can also be accessed via the Figshare

## Abstract

### Background

The costs associated with healthcare are of critical importance to both decision-makers and users, given the limited resources allocated to the health sector. However, the available scientific evidence on healthcare costs in low- and middle-income countries, such as Peru, is scarce. In the Peruvian context, the health system is fragmented, and the private health insurance and its financing models have received less research attention. We aimed to analyse user cost-sharing and associated factors within the private healthcare system.

### Methods

Our study was cross-sectional, using open data from the Electronic Transaction Model of Standardized Billing Data—TEDEF-SUSALUD, between 2021–2022. Our unit of analysis is the user's medical bills. We considered the total amount of cost-sharing, proportion of total payments as cost-sharing, and cost-sharing as a proportion of minimum salaries. We use a multiple regression model to perform the analyses.

### Results

Our study included 5,286,556 health services provided to users of the private health insurance in Peru. We found a significant difference was observed in the cost-sharing for hospitalization-related services, with an average of 419.64 soles per day (95% CI: 413.44 to 425.85). Also, we identified that for hospitalization-related services per day is, on average, 0.41 (95% CI: 0.41 to 0.41) minimum salaries more expensive than outpatient care, although cost-sharing per day of hospitalization represent on average only 14% of the total amount submitted.

### Conclusions

Our study provides a detailed overview of cost-sharing in the private healthcare system in Peru and the factors associated with them. Policymakers can use the study's finding that higher cost-sharing for inpatient hospitalization compared to outpatient care in private

repository (Link: https://doi.org/10.6084/m9.figshare.23932290).

**Funding:** This research was supported by the Dirección de la Universidad Peruana de Ciencias Aplicadas through the UPC-EXPOST-2024-2 incentive. We extend our gratitude to the Dirección de Investigación of the Universidad Peruana de Ciencias Aplicadas for their support in the completion of this research work.

**Competing interests:** The authors have declared that no competing interests exist.

insurance can create inequities in access to healthcare to design policies aimed at reducing these costs and promoting a more equitable and accessible healthcare system in Peru.

## Background

The cost of healthcare is a key issue in resource allocation for both decision-makers and users [1]. Several factors influence the variation in total treatment costs, including age, gender, comorbidities, treatment characteristics, complications and hospital characteristics [2]. In addition, the final price that users pay for health services varies according to the specific services and treatments received [3]. Professional, pharmaceutical, hospital and outpatient services are the main contributors to the average cost of healthcare [4]. Variability in final prices can affect access to healthcare, particularly for people with limited resources, who may find it difficult to meet health insurance cost-sharing [5].

Access to private health insurance varies widely in many low- and middle-income countries. A study of 100 low- and middle-income countries found that average health insurance coverage was 7.9% in low-income countries, 27.3% in lower-middle-income countries, and 52.5% in upper-middle-income countries [6]. Furthermore, within these countries, access to health insurance depends on the economic level of the individual; for example, studies conducted in Ghana found that only a small percentage of the population had access to private health insurance, and these tended to be concentrated in wealthier urban areas [7, 8]. However, we did not identify any previous studies in low- and middle-income countries that assessed whether there were differences in cost-sharing by gender, age group, or other sociodemographic factors.

The scientific evidence on the cost of healthcare is reported mainly for high-income countries, such as the United States, where it is estimated that one in four low-income people could not access a medical consultation because they did not have sufficient financial resources [9]. Similarly, the economic capacity of individuals is a limiting factor for access to healthcare in Colombia, in addition to the lack of employment and the high cost of care, which has an impact on chronicity and health complications, and ultimately on mortality [10]. In this context, the final cost of healthcare in Latin American countries is influenced by several aspects, the most important of which are out-of-pocket spending, privatisation mechanisms, decentralisation, system fragmentation, sources of financing and the type of health insurance coverage available [11]. Although out-of-pocket health spending in Peru decreased between 2008 (US $205.8) and 2017 (US$158.7), economic inequality did not improve significantly [12].

Unlike public health insurance in Peru, which has been the subject of most research, there is limited information on the behaviour of private insurance and its different financing models. Private insurers represent a relatively small proportion of the Peruvian health system (approximately 2.9%) [13] and have received less attention in studies and analyses. In the country, the Health Insurance Fund Administrators Institutions (IAFAS, for its acronym in Spanish) finance care through different mechanisms such as insurance premiums, cost-sharing, and deductibles, among others [14], which implies that users assume a considerable proportion of the costs of care. Unlike care financed by public health insurance in Peru, in private health insurance, patients bear the costs of care, which are the least studied in the country. Cost-sharing is the portion of the total cost of medical care that users must pay out of their pocket when they use their health insurance services [15].

Our study focused on assessing the financial burden of cost-sharing for users and identifying the factors associated with cost-sharing within the private healthcare system in Peru. We

used indicators related to cost-sharing, such as cost-sharing amount, total amount submitted, cost-sharing ratio, and cost-sharing expressed in minimum salaries. The detailed analysis of these indicators allowed us to obtain a better understanding of the cost-sharing assumed by users and its magnitude. In addition, through these indicators, we were able to identify the possible inequities that exist in relation to cost-sharing in the Peruvian private healthcare system.

## Methods

### Design

The study design is cross-sectional and consists of a secondary analysis based on the data available in the open-access database called "Modelo de Transacción Electrónica de datos Estandarizados de Facturación—TEDEF" of the National Superintendence of Health of Peru (SUSALUD, for its acronym in Spanish). The study collects data on invoices settled in the period between January 1, 2021 and December 31, 2022.

### Setting

The Peruvian health system, as mentioned, is fractioned, since it does not depend exclusively on the Ministry of Health, but also on EsSalud, which is administered by the Ministry of Labor, as well as on the Ministry of the Interior through the institutions that manage the Health Insurance Funds (lAFAS) of the Armed Forces, such as the Health Fund of the Peruvian Air Force (FOSFAP), the Health Fund for the Military Personnel of the Peruvian Army (FOSPEME), the Health Fund for the Military Personnel of the Peruvian Navy (FOSMAR) and the Health Fund for the Personnel of the Peruvian National Police, the Regional Governments of the Private Sector with the Health Provider Entities (EPS), the Prepaid IAFAS, i.e. Entities that provide various health plans with medical assistance benefits, the Self-Insurance Companies and the Insurance Companies.

SUSALUD has a crucial role in guaranteeing the rights of all Peruvians to health services by monitoring and enforcing regulations between health service providers (IPRESS) and health insurance companies (IAFAS), which include both the public and private sectors. Therefore, SUSALUD collects information on payments, cost-sharing, and other economic information from some parts of the Peruvian healthcare system and consolidates it in the TEDEF-SUSALUD and other datasets. However, Peru has a fragmented healthcare system, and the existing subsystems may be part of the public and private sectors, creating inequities in both financing and service provision (Social Health Insurance, Ministry of Health, Armed Forces and Police, and private) [16].

It is estimated that around 8.5% of all Peruvians have private health insurance [17], 70% have public health insurance, such as SIS, and 29.8% have Social Health Insurance, which in turn does not prevent an insured person from having parallel private insurance [13]. Similarly, in February 2023, the number of users affiliated with private healthcare providers (EPS) was 2,883,873, of which 33.4% were affiliated to Pacífico EPS S.A., 24.2% to Rímac Perú S.A. EPS, 19.9% to Mapfre Peru S.A. EPS, 12.3% to La Positiva S.A. EPS and 10.2% to Sanitas Perú S.A. EPS [18].

In the private health insurance, no informal payment is recorded. Still, users make different types of expenses, which can be a partial payment of the total cost or the total amount.

### Participants

The unit of analysis is the benefits provided to users of the private health insurance in Peru reported to SUSALUD by the Health Insurance Fund Management Institutions (IAFAS)

through the TEDEF-SUSALUD system. The study included all health services provided to users of any age range, both sexes and from all over Peru. Sampling was not applied as all services financed and approved by IAFAS during the period under review were evaluated.

Inclusion criteria were that participants were between 0 and 100 years of age, that sociodemographic variables were complete, and that benefits were based on a "fee-for-service" mechanism. Other payment modalities, such as patient month, surgical bundles, capitation, and fee-for-service, were excluded because these payment mechanisms bias the relationship between the cost-sharing amount and the cost of the service provided. In addition, health benefits in which cost-sharing were not applied (emergency care, alternative medicine, staff physician and others) and types of affiliation in which cost-sharing are not applied, such as supplementary risk work insurance (SCTR), independent SCTR and compulsory traffic accident insurance (SOAT), were excluded.

## Variables

We define the cost-sharingamount as the amount the member (insured or beneficiary) pays for healthcare received under health insurance coverage. In other words, it is the amount by which the insured contributes to the cost of the healthcare received [19]. In the Peruvian private health insurance system, the quantification of the cost-sharing is based on two components: a fixed criterion called the "deductible", which represents the amount of money paid by the insured before the insurance begins to cover healthcare expenses; and a variable value corresponding to a proportion of the healthcare expenses.

These fixed and variable elements determine the cost-sharing that will be borne by the insured are defined in the insurance plan of each insured person so that the amounts or values of the cost-sharing vary depending on their application to the rates of the various resources consumed in healthcare. In this sense, the value of the cost-sharing is determined by the resources consumed in the treatment of a health problem, including those related to the population treated, aspects of the provision of health services (complexity of the health facility, availability of services and treatments), the type of health service provided to the patient and the medical practice that determines which resources are consumed and the amount of these resources at the preventive, diagnostic, therapeutic, medical or surgical level, among others.

**Total amount paid.**　Total amount paid was defined as the sum of fixed taxable cost-sharing, variable taxable cost-sharing, fixed tax-exempt cost-sharing, and variable tax-exempt cost-sharing. The variable is continuous and was calculated in nuevos soles, which is the Peruvian currency; the average exchange rate to dollars was 3.64 soles per US dollar.

**Total expense presented.**　This is the total amount presented in the invoice reported to TEDEF-SUSALUD. The variable is continuous and was calculated in nuevos soles, the average exchange rate to dollars was 3.64 soles per U.S. dollar.

**Cost-sharing percentage.**　We calculated this as the fraction of the total expenditure that composed the total cost-sharing amount and the total expenditure expressed as a percentage between 0 and 100, that is, the proportion that the user co-pays of the total expenditure.

**Cost-sharing amount in minimum salaries.**　Finally, to understand health expenditure burden, we re-expressed the amount of the cost-sharing made for the service divided by the cost of a minimum wage (1025 soles, for Peru for the year 2022). The minimum wage approximation makes it possible to compare the amount of the cost-sharing with respect to a reference income level in the country and is frequently used in health economics studies [16].

**Covariates.**　The exposure variables for the study were sex (male/female) and age groups according to life cycle: early childhood (0–5 years), infancy (6–11 years), adolescence (12–17 years), youth (18–26 years), adulthood (27–59 years) and old age (60 years or more). In

addition, the type of coverage defined as extra hospital (transfer from the private clinic to the patient's home and recuperative care activities at the patient's home), outpatient, and hospital per day (total amount spent during hospital stay divided by the number of days) was considered. The type of affiliation (regular, optional-independent and complementary) was also evaluated.

### Procedures

The data were obtained from the open-access repository provided by SUSALUD on users who attended in the private health insurance in the years 2021 and 2022. User data are anonymized, so it is not known to whom they are referred. Duplicate or inconsistent data were eliminated.

### Analysis

First, we carried out a descriptive analysis of the socio-demographic characteristics of users who received benefits, with measures of dispersion and central tendency. Second, because our response variables were continuous (total cost-sharing, total amount billed, cost-sharing percentage, and cost-sharing expressed in minimum wages), a generalised linear model was used to assess whether there were associated factors, with gamma distribution for errors. A crude model was presented for each exposure individually, and an adjusted model for all exposures together. The coefficient (beta) was interpreted with 95% confidence intervals, and analyses with $p < 0.05$ were considered significant. In the adjusted analysis for cost-sharing percentage and cost-sharing amount in minimum wages, the generalised linear model with gamma distribution did not converge, so a Gaussian model was chosen. Data analysis was performed using Stata® version 17.0 software (STATA Corporation, College Station, TX, USA).

### Ethical issues

This research used the freely accessible TEDEF database available on the SUSALUD open data platform and the next link: https://doi.org/10.6084/m9.figshare.23932290. It should be noted that the data are available in an anonymized form and do not allow the identification of users. Therefore, the approval of an ethics committee was not necessary.

## Results

### Population characteristics

A total of 10,466,442 records were initially obtained from the TEDEF-SUSALUD database. After a validation process to eliminate duplicate and inconsistent records (n = 1922), as well as those with missing data or payment modalities other than fee-for-service (n = 5,177,964), our study included a final dataset of 5,286,556 health services provided to users of the private health insurance in Peru. The majority of beneficiaries were women, accounting for 60.1% of the sample. The population consisted mainly of adults (27 to 59 years; 58.1%) and the elderly (60 years or more; 19.6%), with outpatient care being the most common type of service (89.9%). In addition, it was found that 79.9% of the beneficiaries were typically affiliated with the regular type of insurance. Detailed socio-demographic characteristics of the beneficiaries and descriptive measures of the response variables are shown in Table 1.

### Total cost-sharing amount

Sex and age did not explain meaningful variation in the total of the cost-sharing amount; however, the type of coverage shows that hospitalization is on average 419.64 (95% CI: 413.44 to 425.85) soles more expensive in the total cost-sharing amount as opposed to outpatient care.

**Table 1. Sociodemographic characteristics of users of private health care services and mean values of outcomes (n = 5,286,556).**

| | n | % | Total cost-sharing amount | | Total expense presented | | Cost-sharing percentage | | Cost-sharing amount in minimum salaries | |
|---|---|---|---|---|---|---|---|---|---|---|
| | | | Mean | SD | Mean | SD | Mean | SD | Mean | SD |
| Sex | | | | | | | | | | |
| Male | 2,107,417 | 39.9% | 89.7 | 293.0 | 545.6 | 1975.6 | 24.0 | 24.7 | 0.09 | 0.29 |
| Female | 3,179,139 | 60.1% | 83.4 | 241.6 | 492.0 | 1521.3 | 23.5 | 24.0 | 0.08 | 0.24 |
| Age group | | | | | | | | | | |
| Early childhood (0–5 years) | 319,035 | 6.0% | 63.5 | 160.5 | 247.8 | 625.0 | 33.2 | 26.1 | 0.06 | 0.16 |
| Infancy (6–11 years) | 259,848 | 4.9% | 65.9 | 139.5 | 304.2 | 643.7 | 27.7 | 24.3 | 0.06 | 0.14 |
| Adolescence (12–17 years) | 216,860 | 4.1% | 73.0 | 202.1 | 357.7 | 921.7 | 26.2 | 24.4 | 0.07 | 0.20 |
| Youth (18–26 years) | 386,439 | 7.3% | 79.6 | 234.2 | 399.9 | 1120.4 | 25.4 | 23.9 | 0.08 | 0.23 |
| Adults (27–59 years) | 3,069,049 | 58.1% | 86.4 | 256.5 | 515.1 | 1676.7 | 23.0 | 24.4 | 0.08 | 0.25 |
| Older (60 years or more) | 1,035,325 | 19.6% | 101.5 | 341.5 | 717.7 | 2401.3 | 20.7 | 22.7 | 0.10 | 0.33 |
| Type of coverage | | | | | | | | | | |
| Extra hospital | 393,833 | 7.5% | 17.5 | 137.2 | 288.5 | 723.9 | 5.2 | 11.1 | 0.02 | 0.13 |
| Outpatient | 4,751,807 | 89.9% | 79.2 | 187.6 | 441.7 | 1379.1 | 25.5 | 24.6 | 0.08 | 0.18 |
| Hospital per day | 140,916 | 2.7% | 504.1 | 1082.7 | 3559.2 | 5951.3 | 14.0 | 14.7 | 0.49 | 1.06 |
| Type of affiliation | | | | | | | | | | |
| Regular | 4,225,966 | 79.9% | 79.1 | 246.1 | 494.3 | 1639.1 | 22.3 | 22.8 | 0.08 | 0.24 |
| Optional (Independent) | 484,587 | 9.2% | 79.3 | 239.3 | 440.8 | 1382.7 | 26.0 | 25.4 | 0.08 | 0.23 |
| Complementary | 576,003 | 10.9% | 141.8 | 374.7 | 714.9 | 2386.0 | 32.1 | 31.3 | 0.14 | 0.37 |
| Total | 5,286,556 | 100% | 86.0 | 263.3 | 513.4 | 1717.1 | 23.7 | 24.3 | 0.08 | 0.26 |

Note: n = number. % = percentage. SD = standard deviation. Values are expressed in nuevos soles, and the average exchange rate to the dollar was 3.64 soles to the US dollar.

In the type of affiliation, it is observed that the total cost-sharing amount in the complementary type affiliations is on average 65.40 (95% CI: 64.64 to 66.16) soles more expensive as opposed to the regular type affiliation (see Table 2).

## Total expense presented

Older adults (60 years or more) and young people (18–26 years) had a higher total expenditure than adults (27–59 years), 359.61 soles (95% CI: 355.7 to 363.53) and 181.62 soles (95% CI: 179.11 to 184.13), respectively. In addition, total expenditure per day of hospitalization was 3185.36 soles (95% CI: 3134.69 to 3236.03) more than outpatient care (see Table 3).

## Cost-sharing percentage

The different types of coverage present a lower proportion of cost-sharing compared to the cost-sharing of outpatient coverage, that is, both in extra hospital (β = -20.65; 95% CI: -20.73 to -20.58) and hospital per day (β = -11.28; 95% CI: -25.28 to -25.04) present a lower proportion of cost-sharing, concerning the total amount. We did not find large differences in the proportion of the cost-sharing amount between sex, age groups or type of affiliation (see Table 4).

## Cost-sharing amount in minimum salaries

Sex, age groups, or type of affiliation did not explain variation in cost-sharing expressed in minimum salary units. However, it was observed that hospital care per day is, on average, 0.41 (95% CI: 0.41 to 0.41) minimum salaries more expensive than outpatient care (see Table 5).

**Table 2. Factors associated with the total amount of cost-sharing in the private health care system in Peru (n = 5,286,556).**

| | | Total cost-sharing | |
|---|---|---|---|
| | | β [CI 95%] | β [CI 95%] |
| | | crude | adjusted |
| Sex | Male | 1 | 1 |
| | Female | **-6.29 [-6.76 to -5.83]** | **-0.33 [-0.43 to -0.22]** |
| Age group | Early childhood (0–5 years) | **-22.95 [-23.67 to -22.23]** | **1.28 [1.08 to 1.48]** |
| | Infancy (6–11 years) | **-20.53 [-21.34 to -19.72]** | **0.52 [0.31 to 0.72]** |
| | Adolescence (12–17 years) | **-13.41 [-14.38 to -12.45]** | **-0.97 [-1.16 to -0.79]** |
| | Youth (18–26 years) | **-6.82 [-7.62 to -6.02]** | **-2.02 [-2.15 to -1.88]** |
| | Adults (27–59 years) | 1 | 1 |
| | Older (60 years or more) | **15.07 [14.42 to 15.72]** | **5.91 [5.64 to 6.18]** |
| Type of coverage | Extra hospital | **-61.74 [-62.02 to -61.46]** | **-65.39 [-65.56 to -65.21]** |
| | Hospital per day | **424.86 [416.64 to 433.08]** | **419.64 [413.44 to 425.85]** |
| | Outpatient | 1 | 1 |
| Type of affiliation | Regular | 1 | 1 |
| | Optional (Independent) | 0.18 [-0.54 to 0.9] | **3.51 [3.34 to 3.68]** |
| | Complementary | **62.71 [61.57 to 63.86]** | **65.40 [64.64 to 66.16]** |

**Note:** The outcome is continuous. Model adjusted by sex, age group, type of coverage, type of affiliation and payment mechanism. Data obtained from the TEDEF-SUSALUD Model (Superintendence Resolution N° 020-2016-SUSALUD/S). Significant values in bold (p<0.05). Values are expressed in nuevos soles, and the average exchange rate to the dollar was 3.64 soles to the US dollar.

**Table 3. Factors associated with the total expenditure presented in the private health care system in Peru. (n = 5,286,556).**

| | | Total expense presented | |
|---|---|---|---|
| | | β [CI 95%] | β [CI 95%] |
| | | crude | adjusted |
| Sexo | Male | 1 | 1 |
| | Female | **-53.62 [-56.64 to -50.59]** | **-24.18 [-25.98 to -22.37]** |
| Age group | Early childhood (0–5 years) | **-267.26 [-270.5 to -264.02]** | **52.61 [49.16 to 56.06]** |
| | Infancy (6–11 years) | **-210.92 [-215 to -206.84]** | **77.04 [73.06 to 81.02]** |
| | Adolescence (12–17 years) | **-157.36 [-162.41 to -152.31]** | **112.90 [109.30 to 116.50]** |
| | Youth (18–26 years) | **-115.15 [-119.49 to -110.81]** | **181.62 [179.11 to 184.13]** |
| | Adults (27–59 years) | 1 | 1 |
| | Older (60 years or more) | **202.59 [197.9 to 207.28]** | **359.61 [355.70 to 363.53]** |
| Type of coverage | Extra hospital | **-153.21 [-156.21 to -150.21]** | **98.76 [96.33 to 101.18]** |
| | Hospital per day | **3117.48 [3060.78 to 3174.18]** | **3185.36 [3134.69 to 3236.03]** |
| | Outpatient | 1 | 1 |
| Type of affiliation | Regular | 1 | 1 |
| | Optional (Independent) | 3.72 [3.65 to 3.8] | **-40.90 [-43.67 to -38.13]** |
| | Complementary | **9.84 [9.76 to 9.93]** | **173.83 [169.46 to 178.21]** |

**Note:** The outcome is continuous. Model adjusted by sex, age group, type of coverage, type of affiliation and payment mechanism. Data obtained from the TEDEF-SUSALUD Model (Superintendence Resolution N° 020-2016-SUSALUD/S). Significant values in bold (p<0.05). Values are expressed in nuevos soles, and the average exchange rate to the dollar was 3.64 soles to the US dollar.

**Table 4. Factors associated with the percentage of cost-sharing, with respect to the expenditure presented, in the private health care system in Peru. (n = 5,286,556).**

| | | Cost-sharing percentage | |
|---|---|---|---|
| | | β [CI 95%] crude | β [CI 95%] ajusted |
| Sex | Male | 1 | 1 |
| | Female | **-0.52 [-0.56 to -0.48]** | **-0.35 [-0.39 to -0.31]** |
| Age group | Early childhood (0–5 years) | **10.17 [10.05 to 10.29]** | **10.27 [10.18 to 10.35]** |
| | Infancy (6–11 years) | **4.67 [4.56 to 4.78]** | **5.02 [4.93 to 5.11]** |
| | Adolescence (12–17 years) | **3.25 [3.13 to 3.36]** | **3.59 [3.49 to 3.69]** |
| | Youth (18–26 years) | **2.45 [2.36 to 2.53]** | **0.69 [0.61 to 0.77]** |
| | Adults (27–59 years) | 1 | 1 |
| | Older (60 years or more) | **-2.34 [-2.39 to -2.29]** | **-3.46 [-3.51 to -3.4]** |
| Type of coverage | Extra hospital | **-20.32 [-20.35 to -20.29]** | **-20.65 [-20.73 to -20.58]** |
| | Hospital per day | **-11.52 [-11.6 to -11.44]** | **-11.28 [-11.4 to -11.15]** |
| | Outpatient | 1 | 1 |
| Type of affiliation | Regular | 1 | 1 |
| | Optional (Independent) | 3.72 [3.65 to 3.8] | **6.28 [6.21 to 6.35]** |
| | Complementary | **9.84 [9.76 to 9.93]** | **10.14 [10.08 to 10.2]** |

**Note:** The outcome is continuous. Model adjusted by sex, age group, type of coverage, type of affiliation and payment mechanism. Data obtained from the TEDEF-SUSALUD Model (Superintendence Resolution N° 020-2016-SUSALUD/S). Significant values in bold (p<0.05). Values are expressed in nuevos soles, and the average exchange rate to the dollar was 3.64 soles to the US dollar.

**Table 5. Factors associated with cost-sharing amount, valued in minimum salaries, in the health benefits of the private health care system in Peru. (n = 5,286,556).**

| | | Cost-sharing amount in minimum salaries | |
|---|---|---|---|
| | | β [CI 95%] crude | β [CI 95%] adjusted |
| Sex | Male | 1 | 1 |
| | Female | **-0.01 [-0.01 to -0.01]** | **-0.01 [-0.01 to -0.01]** |
| Age group | Early childhood (0–5 years) | **-0.02 [-0.02 to -0.02]** | **-0.02 [-0.02 to -0.02]** |
| | Infancy (6–11 years) | **-0.02 [-0.02 to -0.02]** | **-0.01 [-0.02 to -0.01]** |
| | Adolescence (12–17 years) | **-0.01 [-0.01 to -0.01]** | **-0.01 [-0.01 to -0.01]** |
| | Youth (18–26 years) | **-0.01 [-0.01 to -0.01]** | **-0.01 [-0.01 to -0.01]** |
| | Adults (27–59 years) | 1 | 1 |
| | Older (60 years or more) | **0.01 [0.01 to 0.02]** | **0.01 [0.01 to 0.01]** |
| Type of coverage | Extra hospital | **-0.06 [-0.06 to -0.06]** | **-0.06 [-0.06 to -0.06]** |
| | Hospital per day | **0.41 [0.41 to 0.42]** | **0.41 [0.41 to 0.41]** |
| | Outpatient | 1 | 1 |
| Type of affiliation | Regular | 1 | 1 |
| | Optional (Independent) | 0 [0 to 0] | **0.01 [0.01 to 0.01]** |
| | Complementary | **0.06 [0.06 to 0.06]** | **0.06 [0.06 to 0.06]** |

Note: The outcome is continuous. Model adjusted by sex, age group, type of coverage, type of affiliation and payment mechanism. The Peruvian minimum salary for the year 2022 is 1025 soles. Data obtained from the TEDEF-SUSALUD Model (Superintendence Resolution N° 020-2016-SUSALUD/S). Significant values in bold (p<0.05). Values are expressed in nuevos soles, and the average exchange rate to the dollar was 3.64 soles to the US dollar.

## Discussion

Our study found that the average cost-sharing of health benefits in private health insurance in Peru is higher in men than in women, higher in people aged 60 years, or older and in complementary health affiliations; however, the cost-sharing amounts do not show a considerable variation in terms of minimum salaries among the different groups.

On the other hand, we found that there is a considerable difference in the cost-sharing of health benefits when the type of coverage is hospitalization, finding an average cost-sharing per day of hospitalization of 504.1 soles, which corresponds to approximately half a Peruvian minimum monthly salary. Although the cost-sharing per day of hospitalization only correspond to 14% of the total amount submitted on average by a person, this does not limit a person who spends several days hospitalized in a private health centre from incurring catastrophic health expenses. In particular, the Peruvian population perceives that the private healthcare system provides a better quality of service and the time to care is faster than the public healthcare system [20], and healthcare workers consider that the safety culture towards patients is better in private systems [21]. Therefore, individuals seeking faster and better-quality care may incur considerable costs related to their healthcare.

This study makes it possible to identify the average cost-sharing amounts, the expenditures presented, the cost-sharing percentages and the cost-sharing expressed in minimum salaries, which is a potential tool for health decision-making, since it contributes to the estimation of the magnitude of cost-sharing within the private health insurance. It should be noted that, depending on how they are designed, cost-sharing constitute a disincentive and/or incentive mechanism for the use of health services, so it is necessary to have a better understanding of how they affect healthcare seeking behaviour to implement policies or regulatory reforms in this regard.

The introduction of cost-sharing in healthcare systems is intended to regulate expenditure and promote the efficiency of healthcare systems, but it can also lead to inequalities in access to healthcare [22]. Cost-sharing modulates supply and demand, which may affect the choice of services and treatments, especially for people with limited financial resources. Moreover, the introduction of additional cost-sharing could exacerbate existing problems and further disadvantage those who already struggle to pay for healthcare [22].

An assessment of out-of-pocket spending on health found that six out of ten older Peruvians reported spending out-of-pocket on health [23]. This would lead to inequalities in access to health services, especially for vulnerable groups [23]. This study agrees that older adults are one of the age groups with the highest cost-sharing, it is important to recognize that these do not represent the total costs that users may face; therefore, it is very likely that the costs presented in this study underestimate the true financial burden experienced by users. Users may face additional expenses, such as paying for medications, additional testing, transportation, and other services not covered by private insurance. These additional costs can be a significant barrier to accessing care, especially for those with limited financial resources.

To understand private health insurance in Peru, it is important to understand that private insurance users may also have public insurance that covers very complex expenses [24], and that the Peruvian health system is fragmented and insurance has undergone significant historical changes.

### Strengths and limitations

The main strength of the study is that the data are representative of the services provided in private health insurance in Peru. In addition, this study is one of the few that evaluates components of private health spending in Latin America, as well as the approach or study of cost-

sharing in the Peruvian private health insurance. However, our research has limitations that should be mentioned; first, the data do not allow us to clearly describe the medical procedures used, nor the cost of each procedure in detail, nor the complexity of the health services provided; second, it was not possible to clearly evaluate the effect of other variables such as the economic level of the users, the care provided in other public health systems or other information on out-of-pocket spending; finally, this study analyses the cost-sharing of services in the private health insurance, but it was not possible to perform an individual analysis for each user and determine how much each user spent.

## Public health implications

We found that cost-sharing amounts per day for inpatient hospitalization are, on average, 0.41 minimum salaries higher than those for outpatient care. This disparity in cost-sharing can represent a catastrophic financial burden on the health of people who rely on private insurance and require prolonged hospitalization. The affordability of healthcare services is a crucial factor in ensuring equitable access and adequate care for the population. Therefore, effective policies and interventionist measures are required to address this situation and reduce the cost-sharing amount per day of hospitalization. This would help to avoid adverse consequences for people's health and promote greater equity in access to healthcare, thus favouring the general welfare of the population and fostering a fairer and more sustainable health system.

This research has succeeded in making transparent a cost-sharing in private health insurance benefits financed by private insurance and potentially serve for the development of regulations or legislation to promote access to healthcare for privately insured individuals. Therefore, a potential implication is that, although a free market exists in Peru that allows private insurance and clinics to set prices, it is essential to ensure that equity and access to health services is promoted for all Peruvian residents.

## Conclusions

We estimated the cost-sharing and the total expenditure for the services of the private insurance system. On average, a user pays 86 soles per service and the total expenditure for each service is on average 513.4 soles. Furthermore, there is not much difference between men and women, nor between age groups, in terms of the amount paid or spent in total. However, there is a significant difference between the amounts paid and spent for inpatient care compared to outpatient care. In addition, the average user pays half the Peruvian minimum wage per day of hospitalization, which can result in catastrophic health expenditures for people who must spend long periods in the hospital.

## Author Contributions

**Conceptualization:** Luciana Bellido-Boza, David Villarreal-Zegarra.

**Formal analysis:** David Villarreal-Zegarra, Max Pariona-Cárdenas.

**Investigation:** Luciana Bellido-Boza, David Villarreal-Zegarra, Max Pariona-Cárdenas, Roger Carrión, G. J. Melendez-Torres.

**Methodology:** Luciana Bellido-Boza, David Villarreal-Zegarra, Roger Carrión.

**Supervision:** Luciana Bellido-Boza, Roger Carrión, Paul Valdivia-Miranda, G. J. Melendez-Torres.

**Validation:** David Villarreal-Zegarra, Paul Valdivia-Miranda, G. J. Melendez-Torres.

**Visualization:** David Villarreal-Zegarra, Max Pariona-Cárdenas.

**Writing – original draft:** David Villarreal-Zegarra.

**Writing – review & editing:** Luciana Bellido-Boza, Max Pariona-Cárdenas, Roger Carrión, Paul Valdivia-Miranda, G. J. Melendez-Torres.

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
