## [Decision Letter · Decision Letter 0]

10 May 2024

PONE-D-24-01915Co-payments and associated factors in the Peruvian private health care systemPLOS ONE

Dear Dr. Bellido-Boza,

Thank you for submitting your manuscript to PLOS ONE. After careful consideration, we feel that it has merit but does not fully meet PLOS ONE’s publication criteria as it currently stands. Therefore, we invite you to submit a revised version of the manuscript that addresses the points raised during the review process.

Although the study is intriguing, it faces challenges such as the unclear presentation of copayment differences, which must be addressed within the article. Providing a thorough delineation of the Peruvian healthcare system, including both public and private sectors, along with a meticulous exposition of the article's methodological aspects, could greatly enhance clarity.

We look forward to receiving your revised manuscript.

Kind regards,

Oriana Rivera-Lozada de Bonilla

Academic Editor

PLOS ONE

Journal Requirements:

2. Please ensure that you include a title page within your main document. You should list all authors and all affiliations as per our author instructions and clearly indicate the corresponding author.

3. Please include your tables as part of your main manuscript and remove the individual files. Please note that supplementary tables (should remain/ be uploaded) as separate ""supporting information"" files".

Reviewers' comments:

Reviewer's Responses to Questions

**Comments to the Author**

1. Is the manuscript technically sound, and do the data support the conclusions?

Reviewer #1: Partly

Reviewer #2: Yes

2. Has the statistical analysis been performed appropriately and rigorously? 

Reviewer #1: I Don't Know

Reviewer #2: Yes

3. Have the authors made all data underlying the findings in their manuscript fully available?

Reviewer #1: Yes

Reviewer #2: Yes

4. Is the manuscript presented in an intelligible fashion and written in standard English?

Reviewer #1: No

Reviewer #2: Yes

5. Review Comments to the Author

Reviewer #1: Dear Editor-in-chief

Thank you for giving me the opportunity to review the paper entitled” Co-payments and associated factors in the Peruvian private health care system”. It is a cross-sectional analysis using secondary claim data to investigate factors related to the amount of co-payment paid by patient.

My comments follow:

Major comments:

1. I think co-payment is not defined precisely. Co-payments are the flat fee paid by the insured patient each time they access the medical service while co -insurance is fixed percentage that the insured pays. Regarding this definition, co-payment is a fixed amount which is determined by health insurer per service insured gets. It’s normally not different per age or sex ,…..

2. Even if the author means co-insurance as they also have considered the proportion of amount paid by the insured, that also is determined by the health insurance for each type of service and normally it’s not different for age or sex. So I am wondering why the authors expect such difference.

3. It’s not clear if any type of informal payment exists in Peruvian health system or not. If yes, if the dataset includes them or not?

4. I recommend to add a brief explanation about Peruvian health system to the paper. This makes paper easier to be read by an international reader.

5. Private health system and private health insurance are used in the paper interchangeably. I am wondering if these are the same. For instance, isn’t it possible that someone with SHI gets service from private health system?

Reviewer #2: This was an interesting study that viewed the private healthcare insurance co-payment trends and explored possible associated factors in Peru.

In the introduction: Add references to give context in other low and middle income countries regarding private health care insurance, also include references that could show the standard or 'fair' co-payment trends (in proportions or percentage)

In the abstract_ give a brief comment on which type of analysis you used (Linear logistic regression)

In participants section

since you are including all age groups, it's not necessarily an inclusion criteria

In result

Maybe consider using the age group in numbers rather than saying "adults", "young people" or "elderly", it can be a bit confusing.

6. PLOS authors have the option to publish the peer review history of their article (what does this mean?). If published, this will include your full peer review and any attached files.

Reviewer #1: No

Reviewer #2: No

---

## [Author Response · Author response to Decision Letter 0]

30 May 2024

PONE-D-24-01915

Co-payments and associated factors in the Peruvian private health care system

Journal Requirements:

Reply: We confirm that our manuscript conforms to the PlosONE guidelines and journal style.

2. Please ensure that you include a title page within your main document. You should list all authors and all affiliations as per our author instructions and clearly indicate the corresponding author.

Reply: We have added the title to our main document.

3. Please include your tables as part of your main manuscript and remove the individual files. Please note that supplementary tables (should remain/ be uploaded) as separate ""supporting information"" files".

Reply: We confirm that the tables are included in the main manuscript.

Review Comments to the Author

Reviewer #1:

Major comments:

1. I think co-payment is not defined precisely. Co-payments are the flat fee paid by the insured patient each time they access the medical service while co -insurance is fixed percentage that the insured pays. Regarding this definition, co-payment is a fixed amount which is determined by health insurer per service insured gets. It’s normally not different per age or sex ,…..

Even if the author means co-insurance as they also have considered the proportion of amount paid by the insured, that also is determined by the health insurance for each type of service and normally it’s not different for age or sex. So I am wondering why the authors expect such difference.

Reply: The co-payment is understood as the share of health expenditure borne by the insured. For this reason, in the section on variables, the aspects that influence its level (as an expense, not as a criterion or value included in an insurance plan) have been clarified. We add:

“We define the co-payment amount as the amount the member (insured or beneficiary) pays for healthcare received under health insurance coverage. In other words, it is the amount by which the insured contributes to the cost of the healthcare received [19]. In the Peruvian private health insurance system, the quantification of the co-payment is based on two components: a fixed criterion called the "deductible", which represents the amount of money paid by the insured before the insurance begins to cover healthcare expenses; and a variable value corresponding to a proportion of the healthcare expenses. The deductible is paid by the insurer (Supreme Decree No. 008-2010-SA).

These fixed and variable elements of the co-payment are defined in the insurance plan of each insured person so that the amounts or values of the co-payment vary depending on their application to the rates of the various resources consumed in healthcare. In this sense, the value of the co-payment is determined by the resources consumed in the treatment of a health problem, including those related to the population treated, aspects of the provision of health services (complexity of the health facility, availability of services and treatments), the type of health service provided to the patient and the medical practice that determines which resources are consumed and the amount of these resources at the preventive, diagnostic, therapeutic, medical or surgical level, among others..”

2. It’s not clear if any type of informal payment exists in Peruvian health system or not. If yes, if the dataset includes them or not?

Reply: In the Peruvian health system, there is no informal payment that can be recorded, but there are different types of out-of-pocket payments that users make, which can be a partial payment of the total cost or the total amount. If the patient has health insurance, an out-of-pocket payment is made according to the copayment established for the service. The patient may also choose to receive care at full cost. In this context, data on informal payments are not included in the dataset. We add:

“In the Peruvian health system, no informal payment is recorded. Still, users make different types of out-of-pocket expenses, which can be a partial payment of the total cost or the total amount.”

3. I recommend to add a brief explanation about Peruvian health system to the paper. This makes paper easier to be read by an international reader.

Reply: Thanks for the suggestion. We have added a brief explanation of Peru's fragmented health system in the "Setting" section:

"The Peruvian health system, as mentioned, is fractioned, since it does not depend exclusively on the Ministry of Health, but also on EsSalud, which is administered by the Ministry of Labor, as well as on the Ministry of the Interior through the institutions that manage the Health Insurance Funds (lAFAS) of the Armed Forces, such as the Health Fund of the Peruvian Air Force (FOSFAP), the Health Fund for the Military Personnel of the Peruvian Army (FOSPEME),

the Health Fund for the Military Personnel of the Peruvian Navy (FOSMAR) and the Health Fund for the Personnel of the Peruvian National Police, the Regional Governments of the Private Sector with the Health Provider Entities (EPS), the Prepaid IAFAS, i.e. Entities that provide various health plans with medical assistance benefits, the Self-Insurance Companies and the Insurance Companies.”

4. Private health system and private health insurance are used in the paper interchangeably. I am wondering if these are the same. For instance, isn’t it possible that someone with SHI gets service from private health system?

Reply: Thank you for your observation and question. We change "private health system" to "private health insurance" in the manuscript. In this regard, the private health care system in Peru provides services to patients who are financed by private health insurance, to those who pay out-of-pocket in full, and in some cases to patients who are financed by public health insurance; the latter is called a benefit exchange.

Reviewer #2:

5. This was an interesting study that viewed the private healthcare insurance co-payment trends and explored possible associated factors in Peru.

Reply: Thank you for your interest, we will respond to each of your comments below.

6. In the introduction: Add references to give context in other low and middle income countries regarding private health care insurance, also include references that could show the standard or 'fair' co-payment trends (in proportions or percentage)

Reply: We appreciate the reviewer's suggestion to enrich the introduction with references to private health insurance in other low- and middle-income countries and trends in out-of-pocket spending. We have added the following paragraph to the Introduction:

"Access to private health insurance varies widely in many low- and middle-income countries. A study of 100 low- and middle-income countries found that average health insurance coverage was 7.9% in low-income countries, 27.3% in lower-middle-income countries, and 52.5% in upper-middle-income countries [6]. Furthermore, within these countries, access to health insurance depends on the economic level of the individual; for example, studies conducted in Ghana found that only a small percentage of the population had access to private health insurance, and these tended to be concentrated in wealthier urban areas [7, 8]. However, we did not identify any previous studies in low- and middle-income countries that assessed whether there were differences in co-payment by gender, age group, or other sociodemographic factors.”

7. In the abstract_ give a brief comment on which type of analysis you used (Linear logistic regression)

Reply: Thank you for your comment. We add:

“We use a multiple regression model to perform the analyses.”

8. In participants section: Since you are including all age groups, it's not necessarily an inclusion criteria

Reply: Thank you for your comment. We think it is appropriate to point out the inclusion criteria, as there are cases in the TEDEF-SUSALUD database with an age of 101 years and older. Although these cases are very few, we believe that their inclusion could alter our results, as they could be due to recording errors. In addition, our inclusion criterion prevents the inclusion of cases with implausible ages (i.e., 170 years or older), which, although a small proportion, could bias the results. Maintaining this inclusion criterion will allow replication of our findings by limiting them to the 0-100 age group.

9. In result: Maybe consider using the age group in numbers rather than saying "adults", "young people" or "elderly", it can be a bit confusing.

Reply: We agree with the reviewer's suggestion. In the manuscript, we have separated the age groups according to their life cycle:

Early childhood (0-5 years)

Infancy (6 - 11 years)

Adolescence (12 - 17 years)

Youth (18 - 26 years)

Adults (27- 59 years)

Older (60 years or more)

In the results and discussion section, we have placed the appropriate age groups when referring to adults, adolescents, and older adults.

---

## [Decision Letter · Decision Letter 1]

22 Jul 2024

Co-payments and associated factors in the Peruvian private health care system

PONE-D-24-01915R1

Dear Dr.Luciana Elena Bellido-Boza ,

We’re pleased to inform you that your manuscript has been judged scientifically suitable for publication and will be formally accepted for publication once it meets all outstanding technical requirements.

Kind regards,

Oriana Rivera-Lozada de Bonilla

Academic Editor

PLOS ONE

Additional Editor Comments :

**Establish the definition of copayment well in your research**

**Comments to the Author**

1. If the authors have adequately addressed your comments raised in a previous round of review and you feel that this manuscript is now acceptable for publication, you may indicate that here to bypass the “Comments to the Author” section, enter your conflict of interest statement in the “Confidential to Editor” section, and submit your "Accept" recommendation.

Reviewer #1: (No Response)

2. Is the manuscript technically sound, and do the data support the conclusions?

Reviewer #1: Yes

3. Has the statistical analysis been performed appropriately and rigorously? 

Reviewer #1: (No Response)

4. Have the authors made all data underlying the findings in their manuscript fully available?

Reviewer #1: (No Response)

5. Is the manuscript presented in an intelligible fashion and written in standard English?

Reviewer #1: (No Response)

6. Review Comments to the Author

Reviewer #1: I would like to thank the authors for addressing my perevious comments. However, regarding the first comment, I have to mention that the definition provided for the term "co-payment" is actually the definition of user fees and not co-payment.

Cost-sharing or user fees can take the following forms:

Co-payments are a flat fee paid by the insured patient each they access a medical service

A deductible is the amount a person pays before insurance coverage begins to have an effect

Co-insurance is a fixed percentage that the insured pays after the deductible is exceeded, up to the out-of-pocket maximum

Please have a look at Lorna Guinness & Virginia Wiseman, Introduction to Health Economics page 169

7. PLOS authors have the option to publish the peer review history of their article (what does this mean?). If published, this will include your full peer review and any attached files.

Reviewer #1: No

---

## [Editor Report · Acceptance letter]

31 Jul 2024

PONE-D-24-01915R1 

PLOS ONE

Dear Dr. Bellido-Boza, 

I'm pleased to inform you that your manuscript has been deemed suitable for publication in PLOS ONE. Congratulations! Your manuscript is now being handed over to our production team.

Kind regards, 

on behalf of

Dr. Oriana Rivera-Lozada de Bonilla 

Academic Editor

PLOS ONE